# $AT^2ES$: Simultaneous Atmospheric Transmittance-Temperature-Emissivity Separation Using Online Upper Midwave Infrared Hyperspectral Images

**Sungho Kim** [1],*[ID]**, Jungsub Shin** [2] **and Sunho Kim** [2]

1   Department of Electronic Engineering, Yeungnam University, 280 Daehak-Ro, Gyeongsan, Gyeongbuk 38541, Korea
2   Agency for Defense Development, 488-160 Bukyuseong-Daero, Yuseong, Daejeon 34186, Korea; jss@add.re.kr (J.S.); edl423@add.re.kr (S.K.)
*   Correspondence: sunghokim@yu.ac.kr; Tel.: +82-53-810-3530

**Abstract:** This paper presents a novel method for atmospheric transmittance-temperature-emissivity separation ($AT^2ES$) using online midwave infrared hyperspectral images. Conventionally, temperature and emissivity separation (TES) is a well-known problem in the remote sensing domain. However, previous approaches use the atmospheric correction process before TES using MODTRAN in the long wave infrared band. Simultaneous online atmospheric transmittance-temperature-emissivity separation starts with approximation of the radiative transfer equation in the upper midwave infrared band. The highest atmospheric band is used to estimate surface temperature, assuming high emissive materials. The lowest atmospheric band ($CO_2$ absorption band) is used to estimate air temperature. Through onsite hyperspectral data regression, atmospheric transmittance is obtained from the y-intercept, and emissivity is separated using the observed radiance, the separated object temperature, the air temperature, and atmospheric transmittance. The advantage with the proposed method is from being the first attempt at simultaneous $AT^2ES$ and online separation without any prior knowledge and pre-processing. Midwave Fourier transform infrared (FTIR)-based outdoor experimental results validate the feasibility of the proposed $AT^2ES$ method.

**Keywords:** atmospheric transmittance; temperature; emissivity; separation; midwave infrared; hyperspectral images





## 1. Introduction

The concept of temperature and emissivity separation (TES) was originally developed by Gillespie et al. for Advanced Spaceborne Thermal Emission and Reflection Radiometer (ASTER) satellite data analysis [1,2]. Currently, TES is an important research topic in infrared remote sensing applications. Separated temperature can be used to estimate land surface temperature for the study of climate change [3,4]. Emissivity information is useful for mineral composition analysis [5], vegetative cover mapping [6], and object material classification [7].

The scope of this paper is to apply TES online on a flying platform such as an unmanned aerial vehicle. The most critical issue is how to achieve the atmospheric correction to remove the effect of path radiance and atmospheric transmittance in real time without any prior information or pre-processing. Historically, the original TES method in ASTER satellite images used an atmospherically corrected dataset in five multispectral long wave infrared (LWIR) bands [1]. Li et al. compared six methods for extracting relative emissivity spectra from atmospherically corrected multiple spectral bands [3]. Yong et al. tried to estimate atmospheric transmittance in LWIR bands without TES [8]. Payan and Royer further analyzed the applicability and sensitivity of six TES methods [2]. Borel and Tuttle improved TES by using MODTRAN 5-based atmospheric transmittance [9]. Wang et al.

also applied MODTRAN to perform atmospheric correction in a thermal airborne spectrographic imager (TASI) [10]. Adler-Golden et al. adopted simulated atmospheric parameters from the MODTRAN5 model for TES [11]. Wang et al. used the atmospheric transmittance calculated by MODTRAN for TES of Landsat-8 sensor data [12]. Pivovarník et al. improved TES by adopting smoothing in emissivity estimation where atmospheric correction was made using MODTRAN with a mid-latitude summer atmosphere [4].

The previous works have three limitations. First, most require pre-processing of atmospheric correction by using MODTRAN or from prior knowledge. The atmospheric transmittance, downwelling, and upwelling data are generated for TES. Second, TES is conducted offline. Such an approach is impractical for real-time TES on a flying platform because atmospheric conditions change dynamically in time and space. Third, most TES techniques use an LWIR satellite database such as ASTER and TASI.

In this paper, a novel simultaneous atmospheric transmittance-temperature-emissivity separation ($AT^2ES$) method is proposed for online applications based on the following key ideas. First, the radiative transfer equation (RTE) is approximated by considering the physical properties of the upper midwave infrared band (4.2–5.6 μm). Second, the highest and lowest atmospheric transmittance bands are selected. The former is used to estimate surface temperature, and the latter (the $CO_2$ absorption band: 4.2–4.4 μm) is used to estimate air temperature. Through a data regression process, the atmospheric transmittance is estimated with the y-intercept and air temperature. Emissivity is separated using the observed radiance, the separated object temperature, the air temperature, and atmospheric transmittance.

Therefore, the main contributions are summarized as follows.

- The proposed $AT^2ES$ can separate atmospheric transmittance, temperature, and emissivity simultaneously.
- $AT^2ES$ can work online without any prior processing or information.
- $AT^2ES$ can provide a feasible approximate solution in the upper MWIR band (4.2–5.6 μm).

The remainder of this paper is organized as follows. Section 2 explains the proposed $AT^2ES$ method, including the basics of the radiative transfer equation in the upper MWIR band. Section 3 analyzes $AT^2ES$ using a synthetic dataset and outdoor remote sensing data. The paper concludes in Section 4.

## 2. Proposed $AT^2ES$ Method

### 2.1. Basics of the Radiative Transfer Equation

Figure 1 shows hyperspectral imaging in an outdoor environment. It consists of the target, a midwave infrared-Fourier transform infrared (MWIR-FTIR) camera, the sun, and the atmosphere. Observed spectral radiance can be derived from the radiative transfer from Equation (1). Romaniello et al. adopted the radiative transfer equation used in MODTRAN [13]. In general, at-sensor received radiance $L_{obs}(\lambda)$ in the MWIR region consists of opaque object-emitted radiance, reflected downwelling radiance, and total atmospheric path radiance (thermal+solar components).

$$L_{obs}(\lambda) = \tau(\lambda)\left[\varepsilon(\lambda)L_{tg}(\lambda, T_{tg}) + (1 - \varepsilon(\lambda))(L_s^{\downarrow}(\lambda) + L_t^{\downarrow}(\lambda))\right] + L_s^{\uparrow}(\lambda) + L_t^{\uparrow}(\lambda) \quad (1)$$

$L_{obs}(\lambda)$ is the at-sensor radiance; $\lambda$ is the wavelength; $\varepsilon(\lambda)$ is spectral object surface emissivity; $L_{tg}(\lambda, T_{tg})$ is the spectral radiance of the object, assuming a blackbody in the Planck function with object surface temperature $T_{tg}$. $L_s^{\downarrow}(\lambda)$ and $L_t^{\downarrow}(\lambda)$ represent the spectral downwelling solar radiance and thermal irradiance, respectively; $\tau(\lambda)$ is the spectral atmospheric transmittance, and $L_s^{\uparrow}(\lambda)$ and $L_t^{\uparrow}(\lambda)$ are spectral upwelling solar and thermal path radiance, respectively, reaching the sensor. Observed spectral radiance $L_{obs}(\lambda)$ is acquired by applying the Fourier transform to the interferogram in the Michelson interferometer and hot-cold blackbody-based radiometric calibration [14].

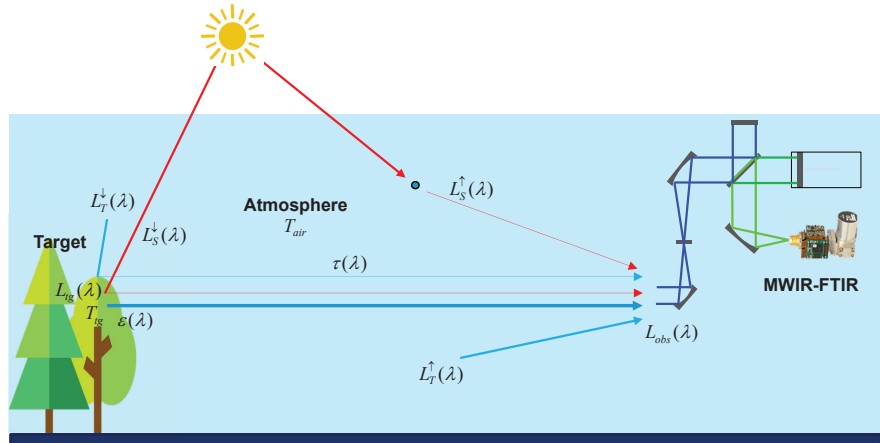

**Figure 1.** Operational concept of $AT^2ES$ using a passive open path Fourier transform infrared imaging system. Notation: $L_s^\downarrow(\lambda)$ and $L_t^\downarrow(\lambda)$ represent the spectral downwelling solar radiance and thermal irradiance, respectively; $L_s^\uparrow(\lambda)$ and $L_t^\uparrow(\lambda)$ are spectral upwelling solar and thermal path radiance, respectively, reaching the sensor.

### 2.2. Proposed Approximation of the RTE in the Upper MWIR Band

Figure 2 visualizes the fractions of total radiance according to the radiometric character-istics for top of atmosphere (TOA): path thermal $L_t^\uparrow(\lambda)$, path reflectance-solar $L_s^\uparrow(\lambda)$, surface reflectance-solar $L_s^\downarrow(\lambda)$, surface reflectance-infrared $L_t^\downarrow(\lambda)$, and surface-emitted $L_{tg}(\lambda, T_{tg})$. The lower MWIR band (3.0–4.2 μm) shows a large fraction for surface reflectance-solar. This means the received radiance strongly depends on the reflected solar energy. However, the contribution of surface reflectance-solar radiance $L_S^\downarrow(\lambda)$ is reduced to only 1% to 4% in the upper MWIR band (4.2–5.6 μm) even for very dry conditions [15]. Figures 3 and 4 show the simulation process of surface reflected-solar and surface emitted-object with the portion of surface reflected-solar. According to the simulation, the average portion of surface reflected solar is 0.65%, which affects negligible error. In addition, surface-reflected down-welling thermal radiance $L_T^\downarrow(\lambda)$ and path reflectance-solar radiance $L_S^\uparrow(\lambda)$ are negligible, compared to surface-emitted radiance $L_{tg}(\lambda, T_{tg})$ and path thermal radiance $L_t^\uparrow(\lambda)$.

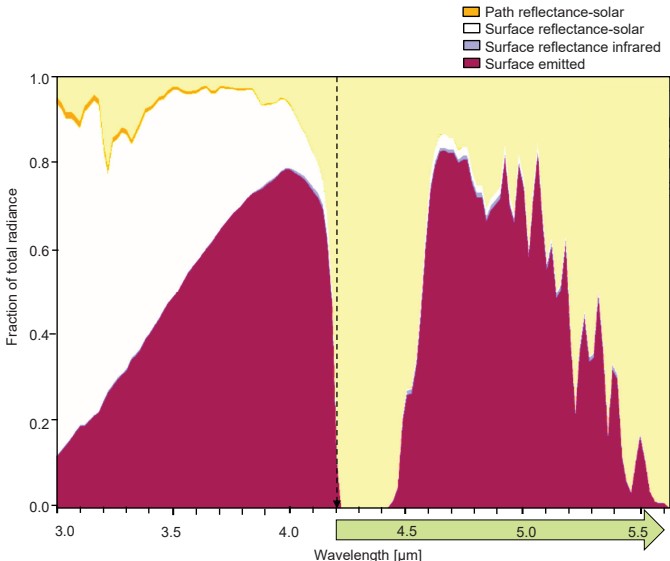

**Figure 2.** Fractional distribution of spectral radiance in the MWIR band.

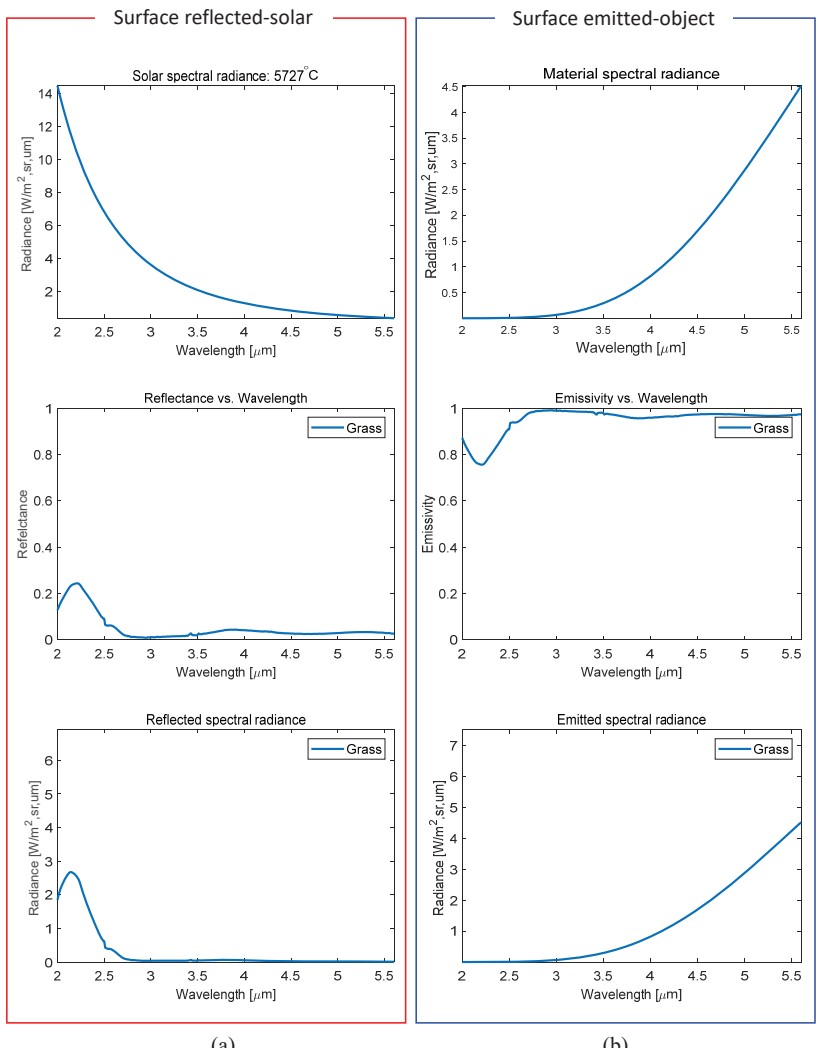

**Figure 3.** (**a**) Generation of surface reflected-solar, (**b**) generation of surface emitted-object. (1st row) solar radiance, object radiance, (2nd row) surface reflectivity, emissivity, and (3rd row) surface reflected-solar, surface emitted-object.

If we ignore surface reflectance-solar, surface reflectance-infrared, path reflectance-solar, we can simplify Equation (1) into Equation (2):

$$L_{obs}(\lambda) = \tau(\lambda)\varepsilon(\lambda)L_{tg}(\lambda, T_{tg}) + L_t^{\uparrow}(\lambda) \tag{2}$$

The definition of thermal upwelling $L_t^{\uparrow}(\lambda)$ is Equation (3) [9]:

$$L_t^{\uparrow}(\lambda) = (1 - \tau(\lambda))L_{BB}(\lambda, T_{air}) \tag{3}$$

where $L_{BB}(\lambda, T_{air})$ denotes the spectral radiance, $[W/(m^2 \cdot sr \cdot \mu m)]$, of a blackbody (Planck's law [16]), and $T_{air}$ is the air temperature in degrees Kelvin [K] of the atmosphere between the object and the camera sensor. The spectral radiation of the atmosphere is modeled as a blackbody [17–19]. Atmospheric path radiance can be described in different ways, but the simplest is to model the particles as blackbodies [19]. $L_{BB}(\lambda, T_{air})$ is defined in Equation (4):

$$L_{BB}(\lambda, T_{air}) = \frac{2hc^2}{\lambda^5(e^{hc/\lambda k T_{air}} - 1)} \tag{4}$$

where *h* denotes Planck's constant, *c* is the speed of light, and *k* is the Boltzmann constant. Therefore, the final form of the proposed approximated RTE is the same as Equation (5):

$$L_{obs}(\lambda) = \tau(\lambda)\varepsilon(\lambda)L_{BB}(\lambda, T_{tg}) + (1 - \tau(\lambda))L_{BB}(\lambda, T_{air}) \tag{5}$$

where $L_{tg}(\lambda, T_{tg})$ was changed to $L_{BB}(\lambda, T_{tg})$ for notational consistency. The proposed RTE is valid for the upper MWIR band (4.2–5.6 μm) with 1–4% radiance uncertainty.

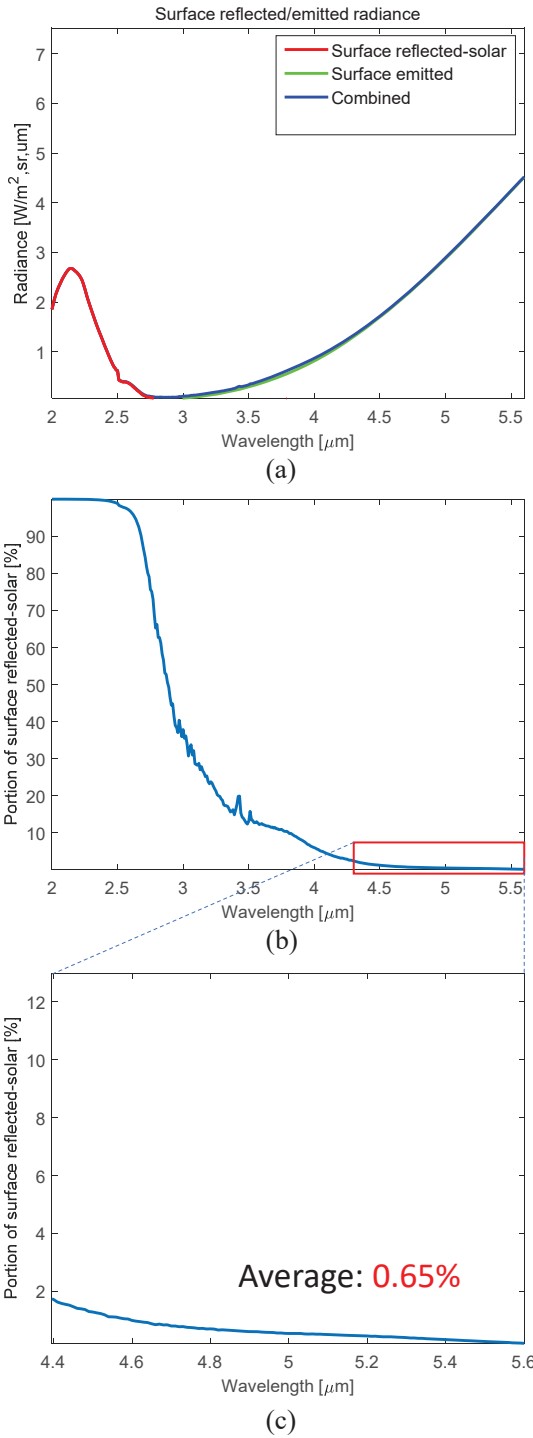

**Figure 4.** Calculation of the portion of surface reflected-solar: (**a**) surface reflected-solar + surface emitted-object, (**b**) portion of surface reflected-solar [%], (**c**) enlarged view in the upper MWIR band.

### 2.3. Details of the $AT^2ES$ Process

Given the approximated RTE model in Equation (5), two unknown temperature parameters ($T_{tg}$, $T_{air}$) should be separated to estimate spectral atmospheric transmittance $\tau(\lambda)$ and spectral emissivity $\epsilon(\lambda)$ given $L_{obs}(\lambda)$. Figure 5 summarizes the overall $AT^2ES$ process, which consists of six blocks: brightness temperature (BT) extraction, $T_{air}$ separation, $T_{tg}$ separation, regression, $\tau(\lambda)$ separation, and $\epsilon(\lambda)$ separation. The BT extraction block converts spectral radiance $L_{obs}(\lambda)$ to brightness temperature units. The band range is limited to the upper MWIR band (4.2–5.6 μm) in order to use the approximate RTE model introduced in the previous subsection. Brightness temperature $BT(\lambda)$ is used in the $T_{air}$ and $T_{tg}$ separation blocks. The regression block estimates slope $a(\lambda)$ and intersect $b(\lambda)$ parameters from observed spectral radiance $L_{obs}(\lambda)$ and target spectral radiance $L_{BB}(\lambda, T_{tg})$. Atmospheric transmittance $\tau(\lambda)$ and target emissivity $\varepsilon(\lambda)$ are separated using these parameters and air radiance $L_{BB}(\lambda, T_{air})$. Each module is explained in the following paragraphs.

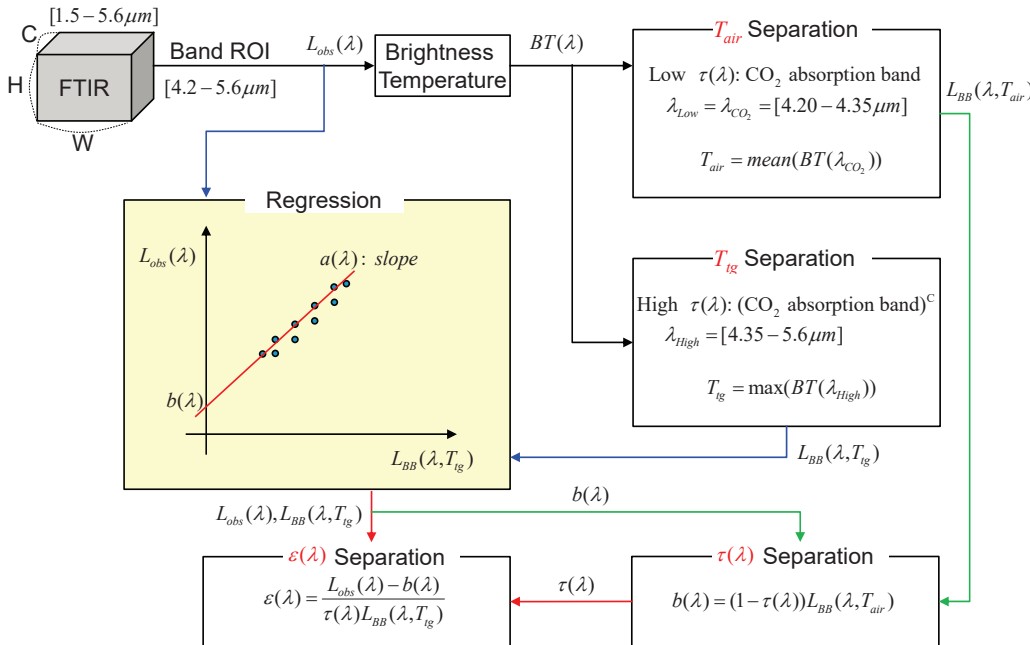

**Figure 5.** Proposed simultaneous $AT^2ES$ flow.

*Brightness temperature module*: The amount of spectral radiance energy can be converted into an equivalent brightness temperature ([20]). By inverting Equation (4), temperature $BT(\lambda)$ [$K$] can be obtained as follows:

$$BT(\lambda) = \frac{hc}{\lambda k \ln(2hc^2/\lambda^5 L_{BB}(\lambda, T) + 1)}. \tag{6}$$

Figure 6 shows an example of brightness temperature extraction from an observed spectral radiance. The remote spectral radiance shows a complicated shape depending on the surface emissivity, atmospheric transmittance, and path radiance. Brightness temperature is the temperature of a blackbody in thermal equilibrium with its surroundings in order to duplicate the observed intensity of a gray-body object at a specific frequency or wavelength. As a result that the spectral radiance provides radiance energy at each wavelength, Equation (6) can calculate the corresponding brightness temperature at each wavelength. Note that a higher brightness temperature can be extracted if atmospheric transmittance and surface emissivity are closer to 1.

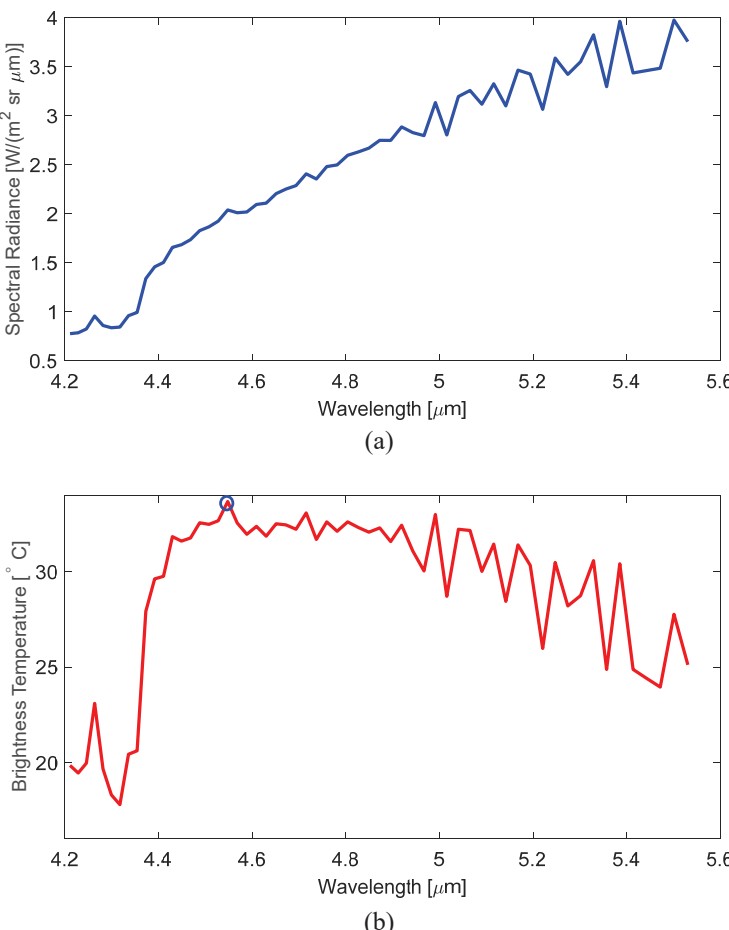

**Figure 6.** Example of brightness temperature extraction from spectral radiance: (**a**) the observed sample spectral radiance [W/(m$^2$ sr μm)], and (**b**) the converted brightness temperature [°C].

$T_{air}$ separation module: According to the MODTRAN simulation in the MWIR band, the spectral transmittance of the carbon dioxide ($CO_2$) band (4.20–4.35 μm) decreases abruptly with distance [21]. The average transmittance in the $CO_2$ band is 0.13, 0.03, 0.005, 0.0001, and 0 at 5 m, 10 m, 20 m, 50 m, and 100 m, respectively. Figure 7 demonstrates the atmospheric transmittance at the 50 m distance in the upper MWIR band. Note that the atmospheric transmittance is 0.0001 in the $CO_2$ absorption band. If we consider only the $CO_2$ band ($\lambda_{CO_2}$ = [4.20–4.35 μm]) with a minimum 20m object distance, transmittance $\tau(\lambda_{CO_2})$ can be regarded as 0, which leads to Equation (7) derived from Equation (5). An MWIR-FTIR camera receives only the upwelling of path thermal radiances in the $\lambda_{CO_2}$ band.

$$L_{obs}(\lambda_{CO_2}) = L_{BB}(\lambda_{CO_2}, T_{air}) \tag{7}$$

Therefore, $T_{air}$ can be obtained by applying a mean operation to Equation (6) in $\lambda_{CO_2}$. The final form of air temperature separation is shown in Equation (8). Figure 8 illustrates an air temperature map image by applying the brightness temperature extraction method to the $CO_2$ absorption band (4.31 μm). A representative air temperature value can be estimated using the spatial and spectral average filter in the $CO_2$ band range.

$$T_{air} = mean(BT(\lambda_{CO_2})) \tag{8}$$

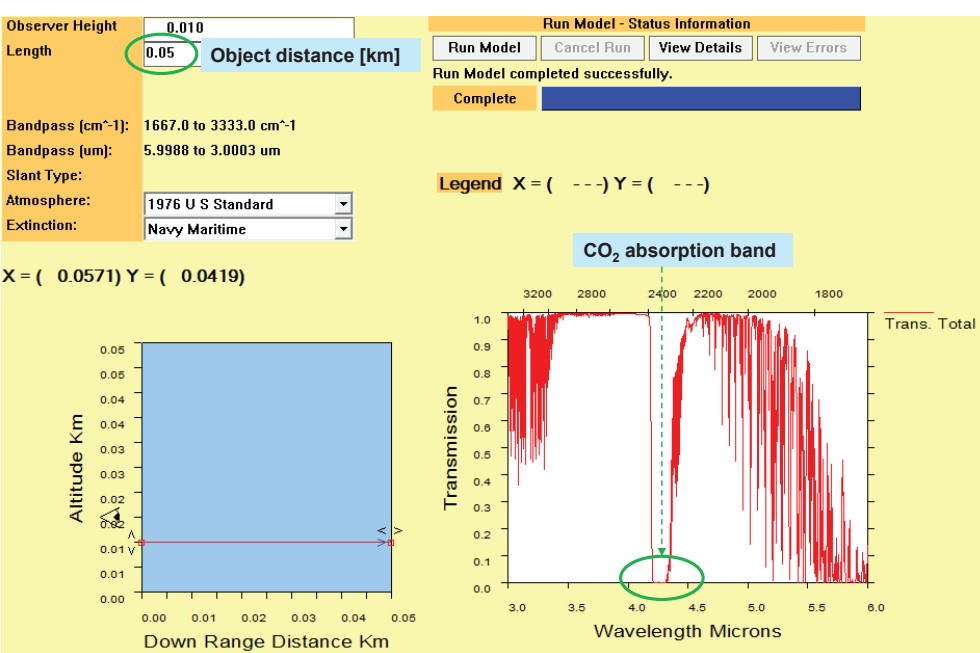

**Figure 7.** Atmospheric transmittance at the 50 m distance, and the characteristics of the $CO_2$ absorption band (4.20–4.35 μm).

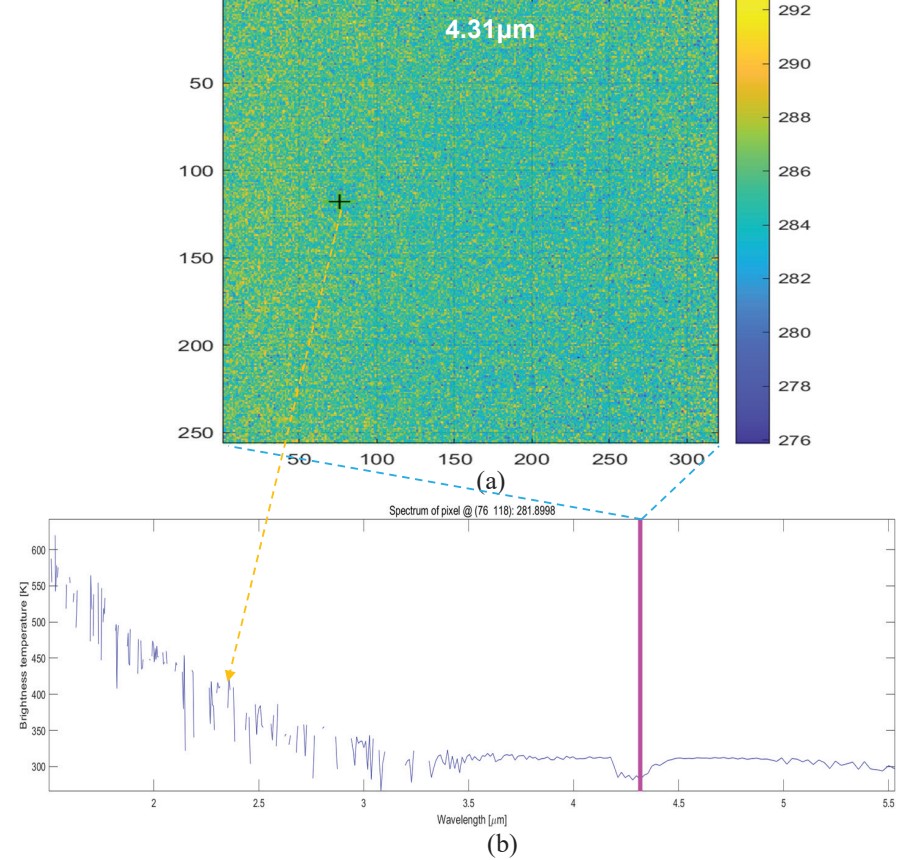

**Figure 8.** Air temperature map extraction using spectral radiance in the $CO_2$ absorption band: (**a**) the air temperature map at 4.31 μm, (**b**) the brightness temperature profile at the cross point in (**a**).

$T_{tg}$ separation module: The remote target temperature separation process requires two assumptions. One is that there must be a high atmospheric transmittance band; the other is that there must be high emissivity band. These assumptions can be satisfied because

the working distance is within 100 m, and most natural and paint materials show high emissivity. Figure 9 proves that maximal transmittance is above 0.992 within a 100 m distance under a clear sky. The average atmospheric transmittance is 0.72 (50 m), 0.66 (100 m), 0.49 (500 m), and 0.41 (1000 m) under 1976 US standard atmosphere model. If the moisture content is 3 times higher (tropical model), the corresponding average atmospheric transmittance is 0.65 (50 m), 0.57 (100 m), 0.39 (500 m), and 0.31 (1000 m). The reduction rate is 13.6% (50 m), 9.7% (100 m), 20.4% (500 m), and 20.4% (1000 m).

The spectral emissivity of the representative materials (paint, grass, asphalt, and concrete) is at least 0.9 as shown in Figure 10.

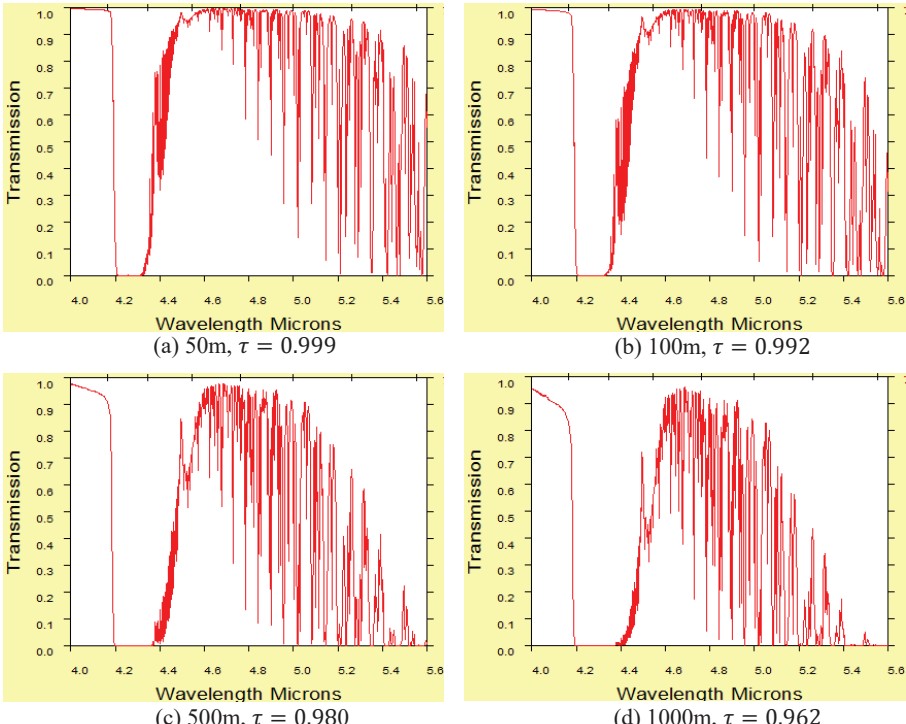

**Figure 9.** Atmospheric transmittance distribution, and the maximum values based on object distance.

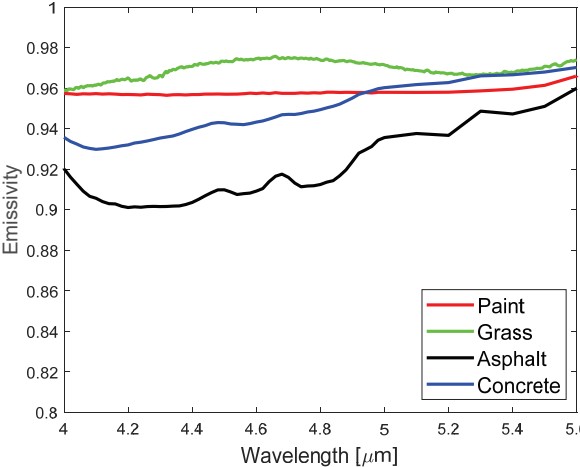

**Figure 10.** Emissivity distributions of various materials in the upper MWIR band.

In these environmental conditions, there is an optimal band with a high maximum $\tau(\lambda_{opt})\varepsilon(\lambda_{opt})$ of 0.9 or more. Therefore, Equation (5) can be reduced to Equation (9) with a maximum 10% margin of error. Target temperature $T_{tg}$ can be obtained by applying brightness temperature to Equation (9). In practical terms, optimal band $\lambda_{opt}$ is unknown a

priori because we have no information on object distances and material types. However, the problem can be bypassed by applying the max operation to Equation (6). The final form of target temperature separation is shown in Equation (10), where $\lambda_{high}$ = [4.35–5.60 μm], which is the complement of the $CO_2$ absorption band. The calculated target temperature (33.6 °C) is the blue circle overlaid in Figure 6.

$$L_{obs}(\lambda_{opt}) = L_{BB}(\lambda_{opt}, T_{tg}) \tag{9}$$

$$T_{tg} = max(BT(\lambda_{high})) \tag{10}$$

Regression module: The proposed approximate RTE, Equation (5), can be written by replacing coefficients as follows:

$$L_{obs}(\lambda) = a(\lambda)L_{BB}(\lambda, T_{tg}) + b(\lambda) \tag{11}$$

where $a(\lambda) = \tau(\lambda)\varepsilon(\lambda)$, and $b(\lambda) = (1 - \tau(\lambda))L_{BB}(\lambda, T_{air})$. Slope $a(\lambda)$ and intercept $b(\lambda)$ can be estimated using regression between $L_{obs}(\lambda)$ and $L_{BB}(\lambda, T_{tg})$, as shown in Figure 11. Hyperspectral data points are obtained from different areas with the same distance. Each observed spectrum provides the BT from which $T_{tg}$ is separated by maximization, as explained above. Figure 12 shows the regressed coefficients for each wavelength.

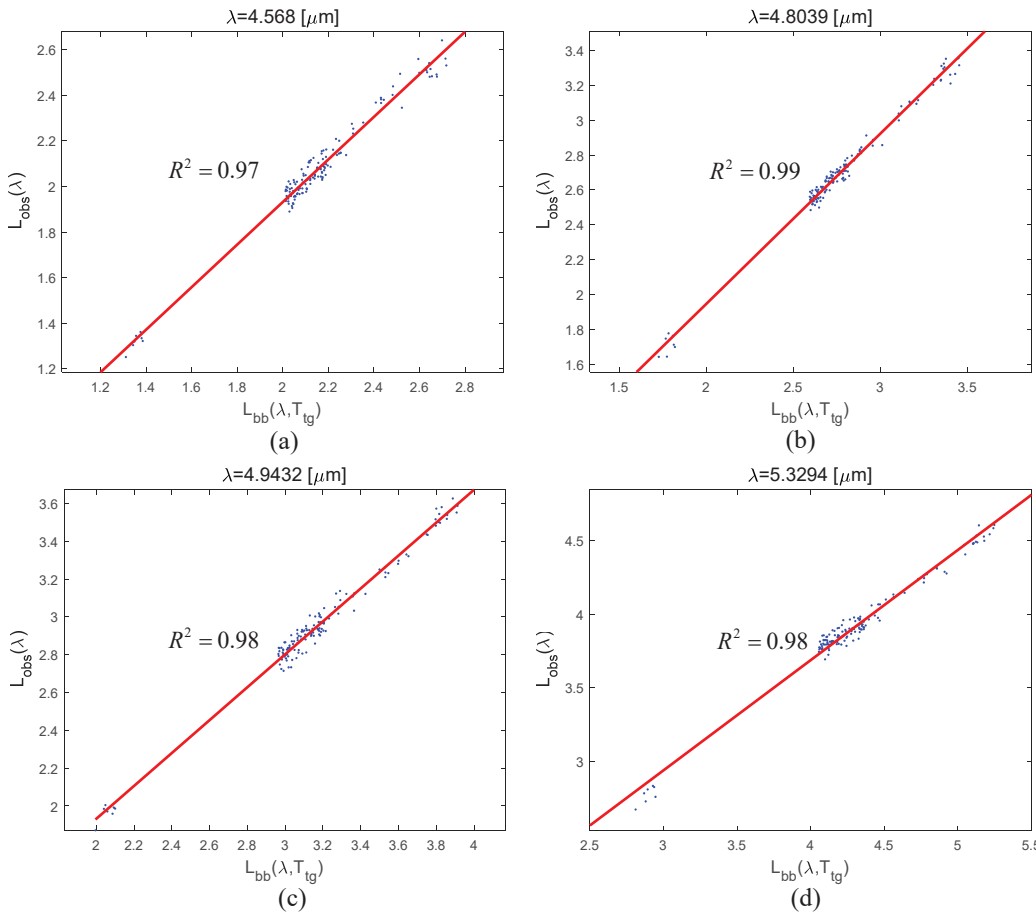

**Figure 11.** Examples of linear regression between $L_{obs}(\lambda)$ and $L_{BB}(\lambda, T_{tg})$ for the representative bands: (**a**) $\lambda = 4.568$[μm], (**b**) $\lambda = 4.8039$[μm], (**c**) $\lambda = 4.9432$[μm], (**d**) $\lambda = 5.3294$[μm].

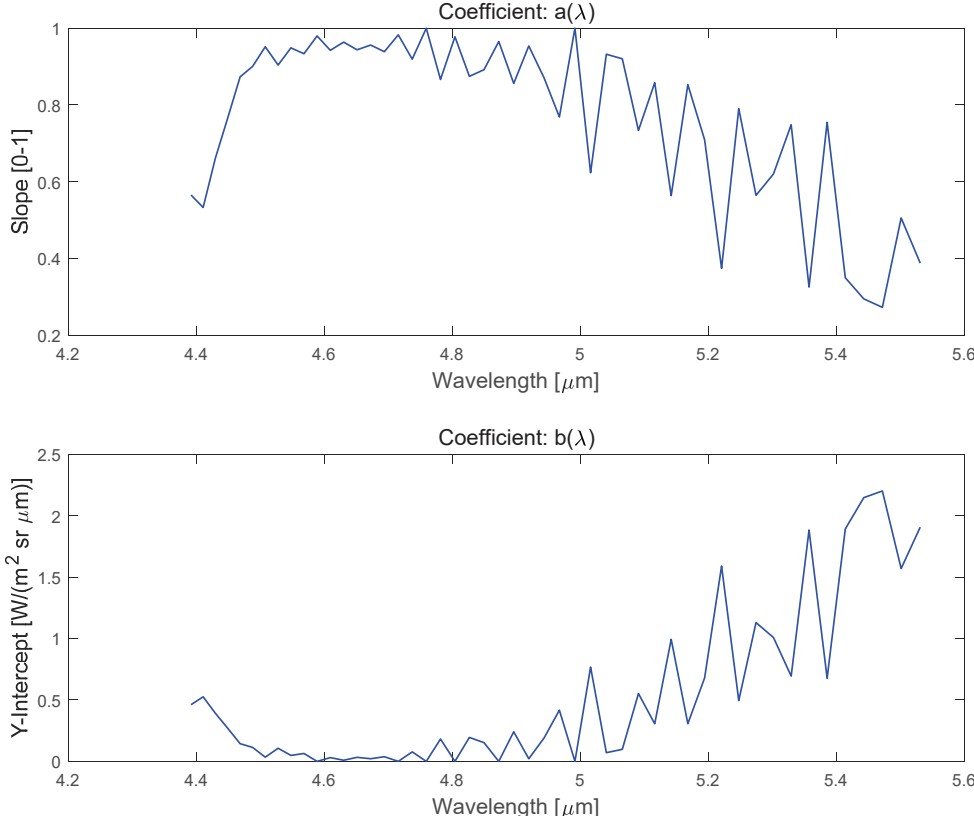

**Figure 12.** Examples of slope $a(\lambda)$ and y-intercept $b(\lambda)$ coefficients in linear regression.

$\tau(\lambda)$ separation module: As a result that $b(\lambda) = (1 - \tau(\lambda))L_{BB}(\lambda, T_{air})$, atmospheric transmittance $\tau(\lambda)$ can be calculated using $b(\lambda)$ and $L_{BB}(\lambda, T_{air})$ as follows:

$$\tau(\lambda) = 1 - \frac{b(\lambda)}{L_{BB}(\lambda, T_{air})} \tag{12}$$

Atmospheric temperature $T_{air}$ provides blackbody radiation, and y-intercept $b(\lambda)$ is separated through linear regression. Figure 13 (top chart) shows an example of separated atmospheric transmittance using Equation (12).

$\varepsilon(\lambda)$ separation module: In Equation (5), spectral emissivity $\varepsilon(\lambda)$ can be separated using atmospheric transmittance $\tau(\lambda)$, object temperature $T_{tg}$, and observed spectral radiance $L_{obs}(\lambda)$, as seen in Equation (13). As a result that each sample has its own spectral emissivity, a representative spectral emissivity profile can be obtained via sample mean. Figure 13 (bottom) shows an example of separated emissivity using Equation (13).

$$\varepsilon(\lambda) = \frac{L_{obs}(\lambda) - b(\lambda)}{\tau(\lambda)L_{BB}(\lambda, T_{tg})} \tag{13}$$

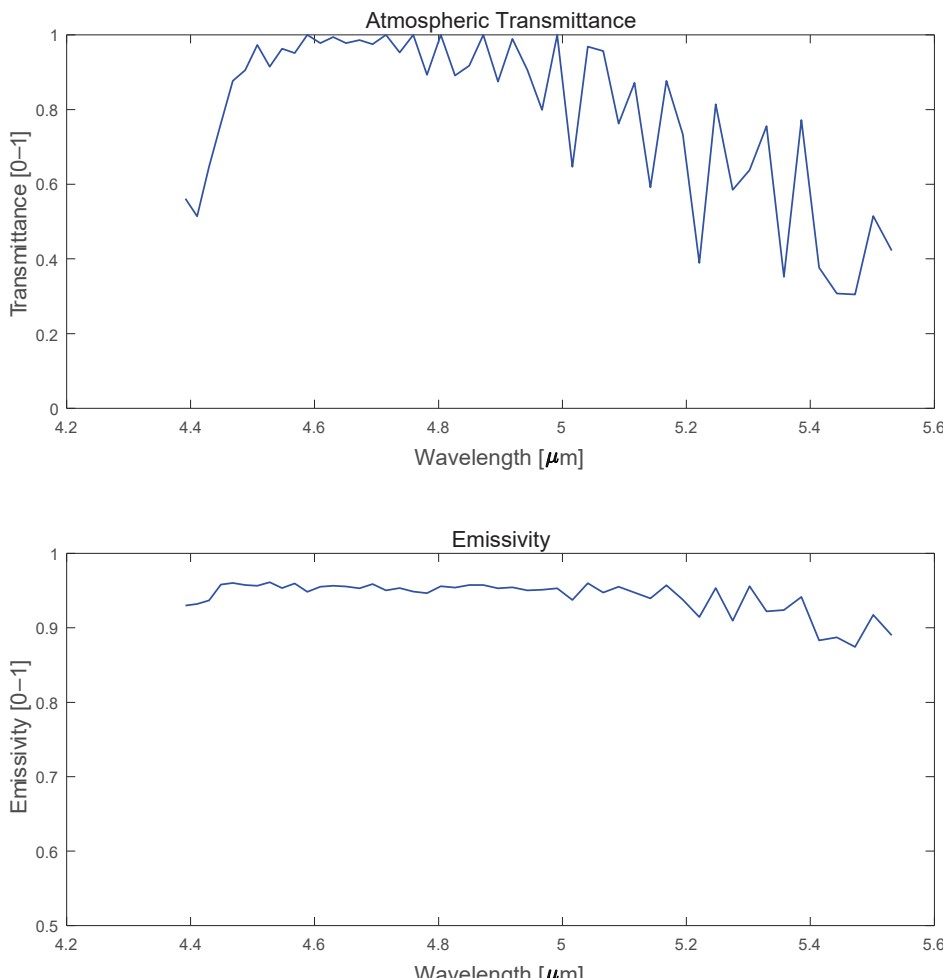

**Figure 13.** Top chart shows separated atmospheric transmittance, and bottom chart, separated emissivity of a sample plane.

## 3. Experimental Results

### 3.1. Experiments Using Synthetic Hyperspectral Datasets

In the first experiment, synthetic hyperspectral data were generated for parameter analysis using Equation (5). The four critical parameters are object temperature $T_{tg}$, air temperature $T_{air}$, emissivity $\varepsilon(\lambda)$, and atmospheric transmittance $\tau(\lambda)$. Figure 14 demonstrates the synthetic spectrum generation flow for observed signal $L_{obs}(\lambda)$. Figure 14a is the grass spectrum downloaded from the ECOSTRESS library, 15 August 2020 (https://ecostress.jpl.nasa.gov/) [22]. Figure 14b presents spectral blackbody radiance of an object with temperature $T_{tg}$ = 30 °C. Figure 14c is emitted object radiance from multiplying Figure 14a,b. The observed spectral radiance in Figure 14f was generated by applying the atmospheric transmittance in Figure 14d to the emitted object radiance and the path radiance in Figure 14e.

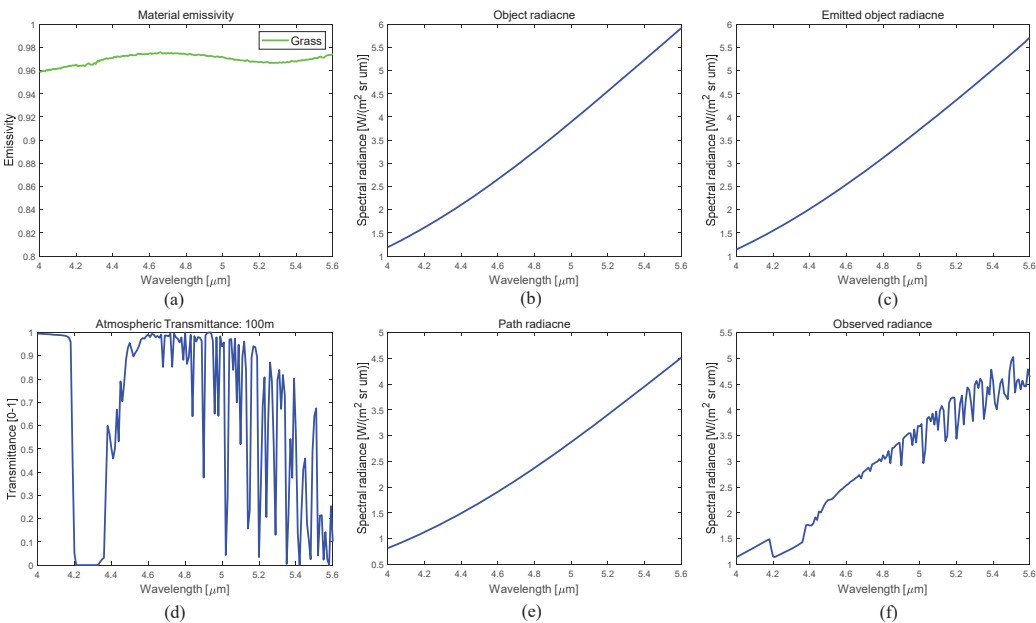

**Figure 14.** Synthetic spectrum generation flow: (**a**) grass emissivity, (**b**) object radiance, (**c**) emitted object radiance, (**d**) atmospheric transmittance, (**e**) path radiance, and (**f**) observed radiance.

Through the generation process, 200 observed spectra were generated, as seen in Figure 15a. As a baseline dataset, Gaussian noise was added with the following parameters: $\sigma_\tau = 0.0001$; $\sigma_{T_{tg}} = 1$; $\sigma_{T_{air}} = 0.0001$, and $\sigma_\varepsilon = 0.0001$, where $\sigma_\tau$ denotes the standard deviation of atmospheric transmittance, $\sigma_{T_{tg}}$ denotes the standard deviation of object temperature, $\sigma_{T_{air}}$ is the standard deviation of air temperature, and $\sigma_\varepsilon$ is the standard deviation of object emissivity. The $\sigma_{T_{air}}$ is set as 0.0001 to consider only the effect of $\sigma_{T_{tg}}$. A value of 0.0001 is the minimal numerical value for simulation purposes. Figure 15b shows an example of a brightness temperature profile converted from an original spectral radiance. The maximum value is regarded as the object temperature, and each separated sample's temperature is displayed in Figure 15c. Each brightness temperature in the $CO_2$ band provides a candidate air temperature, as shown in Figure 15d. The average of the distribution is regarded as the final air temperature. In this baseline dataset, the separated air temperature is 29.99 °C.

Figure 16's left side presents the estimated coefficients of slope and intercept for the upper MWIR band. Figure 16's right side shows an example of linear regression at $\lambda = 5.6$ μm indicating the slope and intercept. Final separation of atmospheric transmittance and emissivity is achieved by applying Equations (12) and (13) to the separated parameters and observed spectrum, as shown in Figure 17. In this case, the mean absolute error (MAE [23]) of spectral atmospheric transmittance is 0.013, and that of spectral emissivity is 0.015.

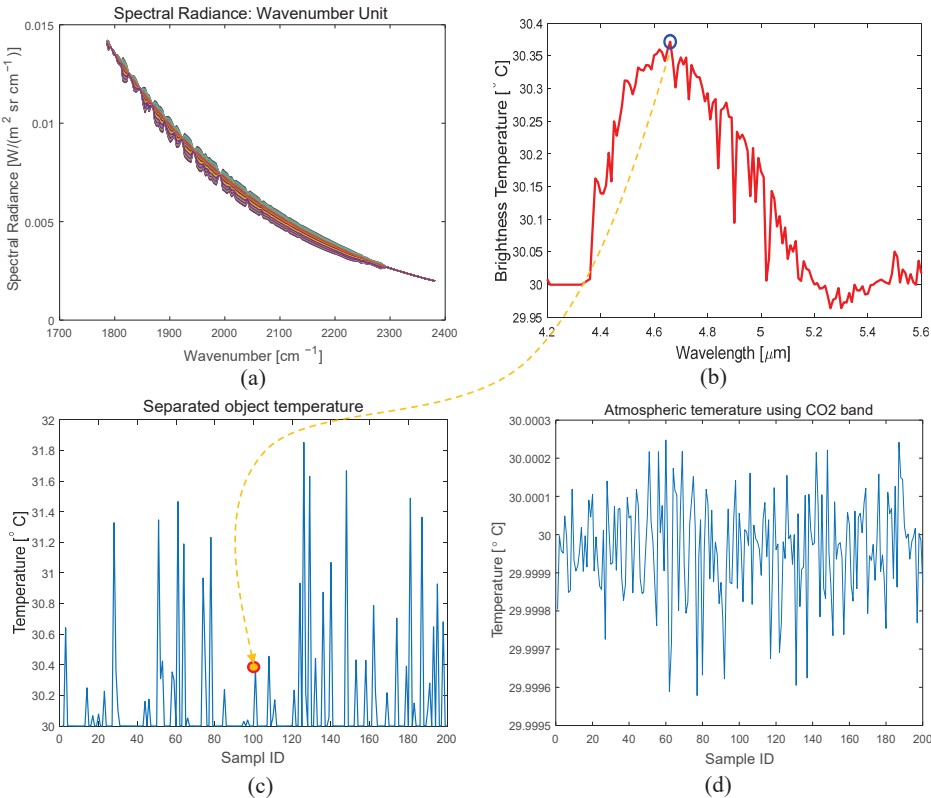

**Figure 15.** Temperature separation from synthetic spectra: (**a**) generated synthetic data (200 spectra), (**b**) brightness temperature and peak value for a sample spectrum, (**c**) the distribution of separated object temperatures, and (**d**) the distribution of separated atmospheric temperatures using the $CO_2$ absorption band.

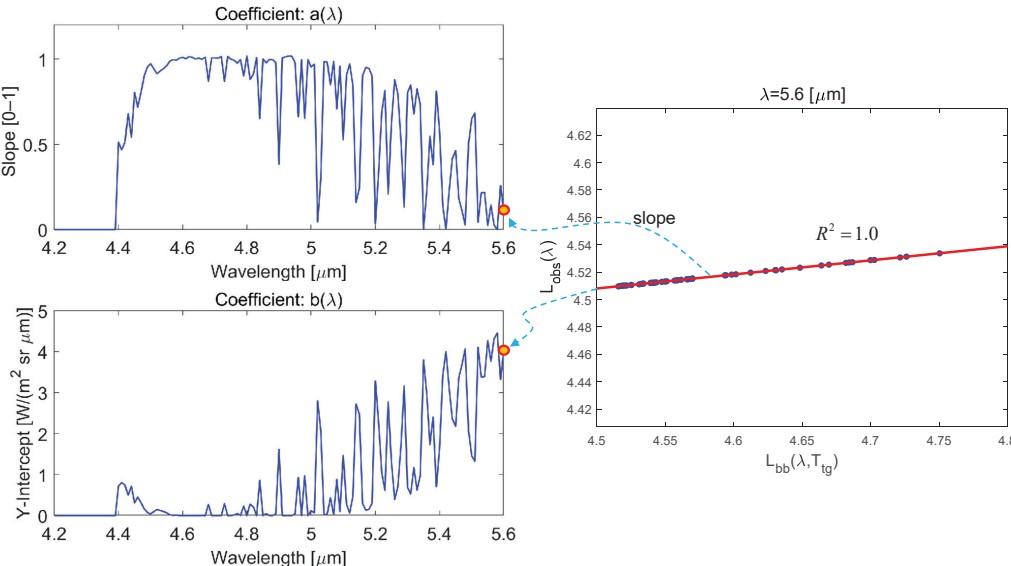

**Figure 16.** Data regression from synthetic spectra: (**left**) slope coefficient and y-intercept coefficient, and (**right**) data regression example for the 5.6 μm wavelength.

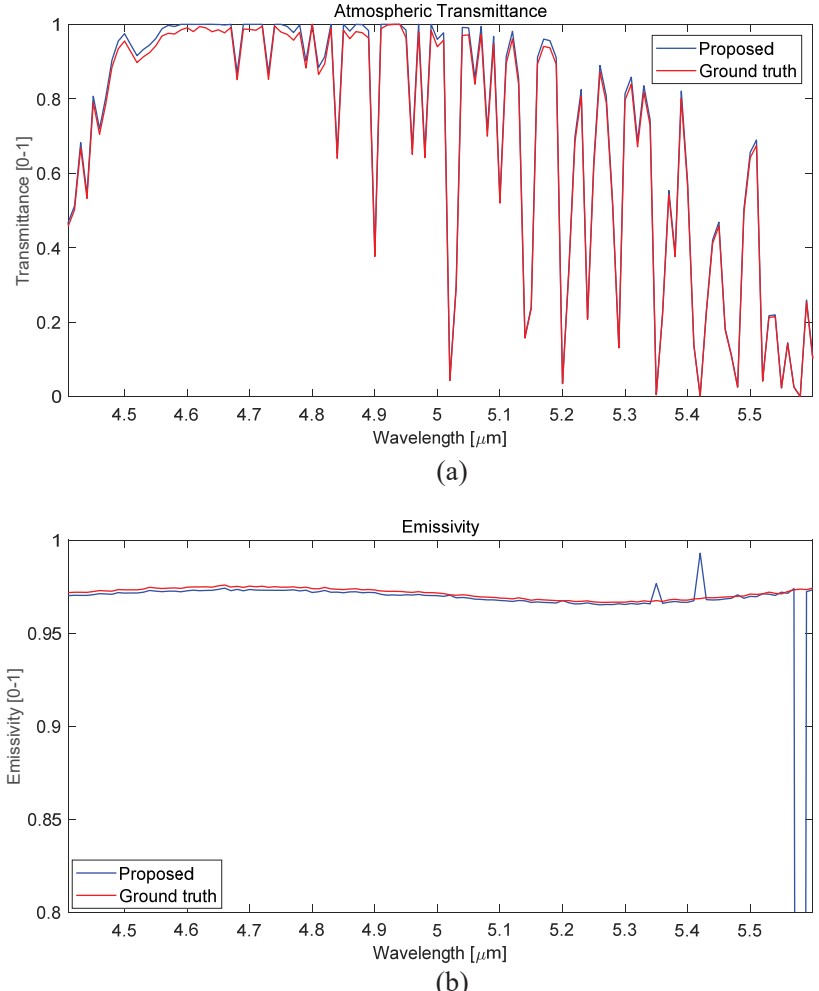

**Figure 17.** Separated atmospheric transmittance and emissivity: (**a**) a comparison of spectral atmospheric transmittance between the proposed method and ground truth, and (**b**) a comparison of spectral emissivity between the proposed method and ground truth.

It is important to analyze the effects of noise in simultaneous four-parameter $(T_{tg}, T_{air}, \tau(\lambda), \varepsilon(\lambda))$ separation. The MAE performance metric is used to check the trend. If $\sigma_{T_{tg}}$ varies from 0.5 to 4.0, the MAEs of the four parameters are shown in Figure 18. As the object surface temperature variation increases, the error in emissivity and air temperature increases. On the other hand, the atmospheric transmittance separation error is reduced.

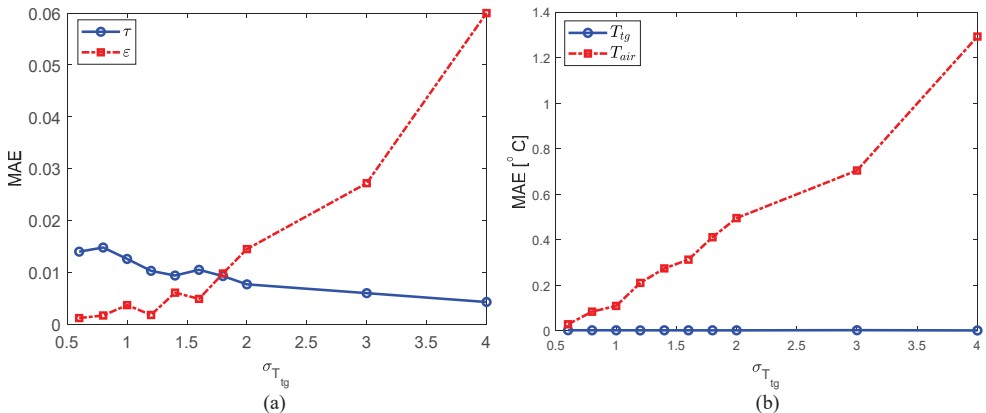

**Figure 18.** Parameter separation performance according to object temperature noise ($\sigma_{T_{tg}}$): (**a**) MAE of $\tau, \varepsilon$, and (**b**) MAE of $T_{tg}, T_{air}$.

If $\sigma_{T_{air}}$ varies from 0.0001 to 2.0, the MAEs of the four parameters are as shown in Figure 19. As the air temperature noise increases, the error in atmospheric transmittance, object temperature, and air temperature increases. On the other hand, emissivity separation error has almost no relation to air temperature noise.

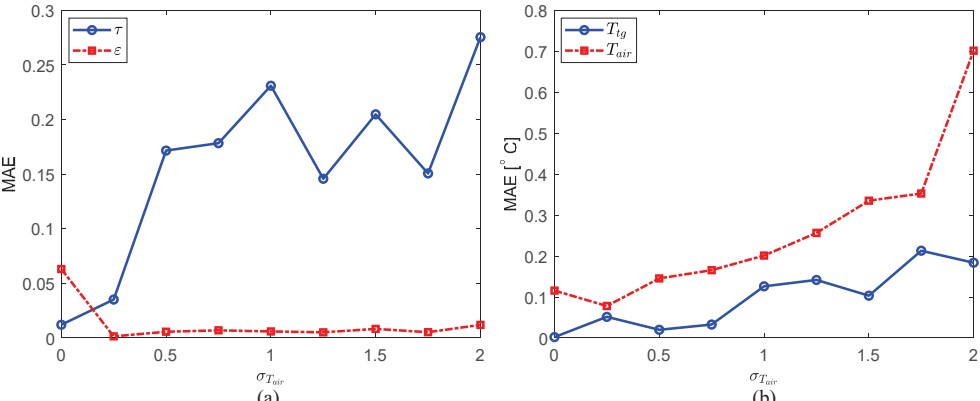

(a)

(b)

**Figure 19.** Parameter separation performance based on air temperature noise ($\sigma_{T_{air}}$): (**a**) MAE of $\tau, \varepsilon$, and (**b**) MAE of $T_{tg}, T_{air}$.

If $\sigma_{\tau}$ varies from 0.0001 to 0.0008, the MAEs of the four parameters are as shown in Figure 20. As the atmospheric transmittance noise increases, the error in atmospheric transmittance increases. On the other hand, other parameter separation errors have almost no relation to atmospheric transmittance noise.

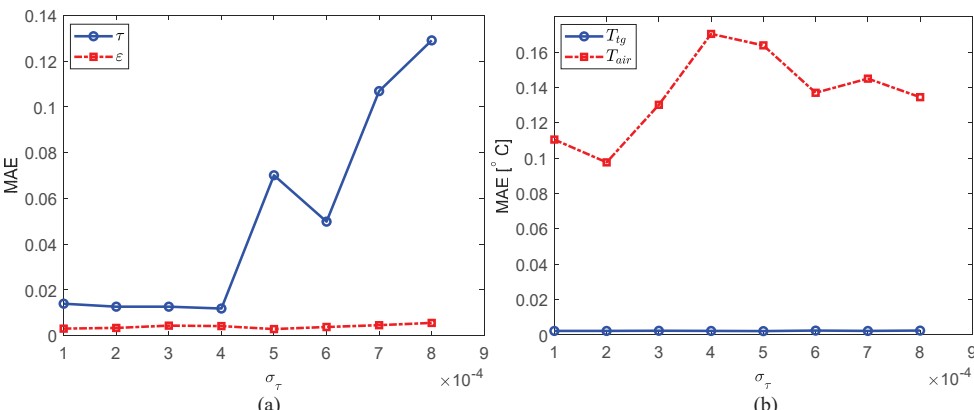

(a)

(b)

**Figure 20.** Parameter separation performance based on atmospheric transmittance noise ($\sigma_{\tau}$): (**a**) MAE of $\tau, \varepsilon$, and (**b**) MAE of $T_{tg}, T_{air}$.

Finally, if $\sigma_{\varepsilon}$ varies from 0.0001 to 0.1, the MAEs of the four parameters are as shown in Figure 21. As emissivity noise increases, the error in atmospheric transmittance and emissivity increases. In a small noise interval (0.0001–0.01), the error in air temperature increases sharply. Object temperature separation errors have almost no relation to emissivity noise.

To verify the approximation of the RTE in Equation (2), the effect of path reflectance-solar in air temperature estimation was conducted as shown in Figure 22a. The portion of path reflectance-solar was varied from 0 to 0.5%. The corresponding temperature error was 0 to 0.138 °C. Likewise, the effect of surface reflectance-infrared in target temperature estimation was conducted as shown in Figure 22b. In this case, the effect is more negligible due to the small reflectivity (0.05 in case of grass) in the upper MWIR band.

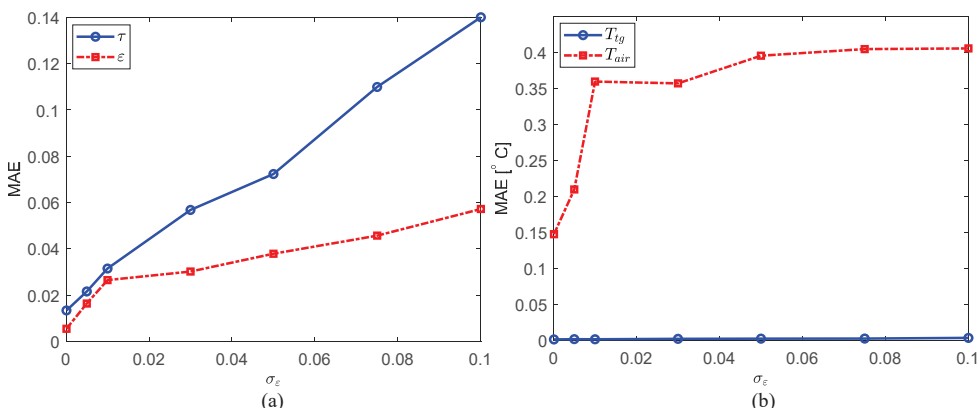

**Figure 21.** Parameter separation performance based on emissivity noise ($\sigma_\varepsilon$): (**a**) MAE of $\tau, \varepsilon$, and (**b**) MAE of $T_{tg}, T_{air}$.

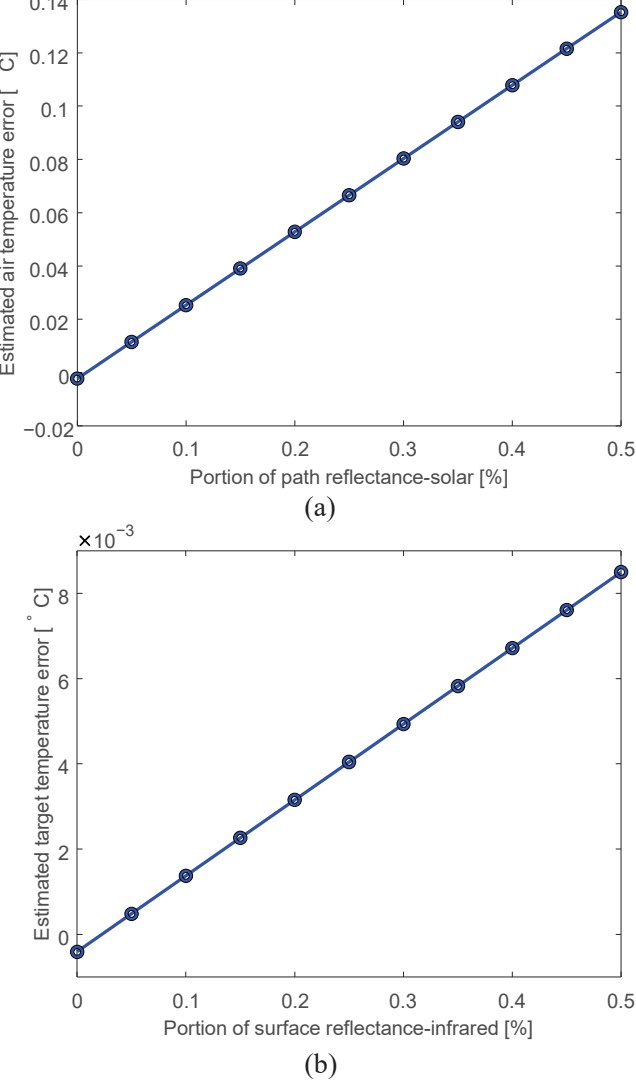

**Figure 22.** (**a**) Effect of path reflectance-solar in air temperature estimation, (**b**) effect of surface reflectance-infrared in target temperature estimation.

### 3.2. Experiments Using Real Hyperspectral Datasets

In the second experiment, the feasibility of the proposed $AT^2ES$ was validated for practical applications. Figure 23 shows the hyperspectral data acquisition environment

and the data sampling points for evaluation. MWIR hyperspectral images were acquired with the Telops Hyper-Cam MWE model [24]. It can provide calibrated spectral radiance images with a high spatial and spectral resolution from a Michelson interferometer in the shortwave to midwave band (1.5–5.6 μm). Spatial image resolution was 320 × 240, with spectral resolution at up to 0.25 cm$^{-1}$. The noise equivalent spectral radiance (NESR) was $7[nW/(cm^2 \cdot sr \cdot cm^{-1})]$, and the radiometric accuracy was approximately 2 K. The field of view was 6.5 × 5.1 deg.

In this paper, only the upper MWIR band (4.2–5.6 μm) was used for our valid approximate RTE model. Although a top-down aerial surveillance scenario is ideal, we chose a ground-based side-looking scenario, because the TELOPS MWE camera is too huge and heavy for an airplane to carry. Note that a narrow horizontal region was selected in order to use the assumption of common atmospheric transmittance. In addition, there were 450 grass samples and 450 asphalt samples.

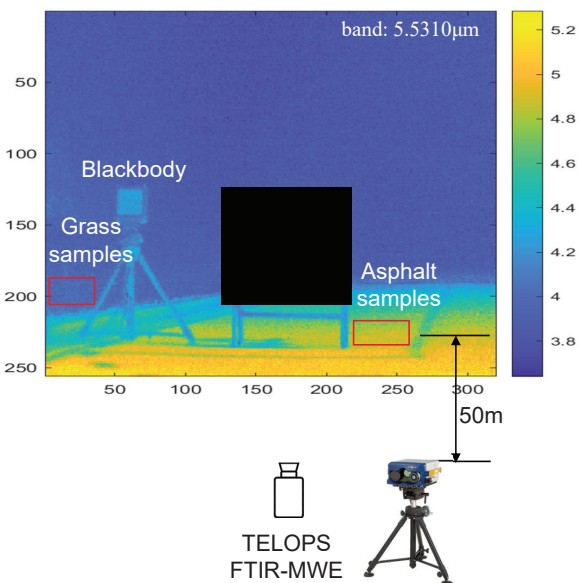

**Figure 23.** Outdoor field test environment and hyperspectral data acquisition scenario.

Our proposed $AT^2ES$ method can simultaneously extract four parameters: $T_{air}, T_{tg}, \tau(\lambda), \varepsilon(\lambda)$. According to the experimental results, the estimated $T_{air}$ was 20.8 °C, which is 0.5 °C lower than the reference air temperature provided by the Korea Meteorological Administration (21.3 °C). In addition, the estimated $T_{tg}$ was 21.8 °C. The ground truth for grass temperature is hard to measure due to weak leaves and complex structures. Normally, grass temperature is almost the same as air temperature in a thermal equilibrium state [25]. In general, grass has high albedo and high emissivity (>0.95). High albedo prevents solar energy absorption and high emissivity absorbs the thermal energy radiated by near air. Under no wind state, the assumption that grass temperature is almost the same as air temperature is reasonable. However, if the wind is strong, the evapotranspiration from the grass is an important factor which led to its lower temperature [25].

Figure 24 shows the estimated spectral atmospheric transmittance and emissivity, compared with MODTRAN and the ECOSTRESS grass library. In the MODTRAN simulation, object distance was set to 50 m in a mid-latitude spring environment. Note that $AT^2ES$ can estimate spectral atmospheric transmittance quite accurately, as shown in Figure 24a. In the emissivity comparison, sample No. VH351 (Bromus diandrus) from the ECOSTRESS spectral library was chosen because it was most similar to our grass region. Considering the complex grass composition, $AT^2ES$ estimated a similar emissivity profile, as shown in Figure 24b. Figure 25 visualizes the spectral estimation error of $\tau(\lambda)$, $\varepsilon(\lambda)$. The MAEs of atmospheric transmittance and emissivity were 0.087 and 0.063, respectively. Note that large errors were generated around low atmospheric transmittance bands.

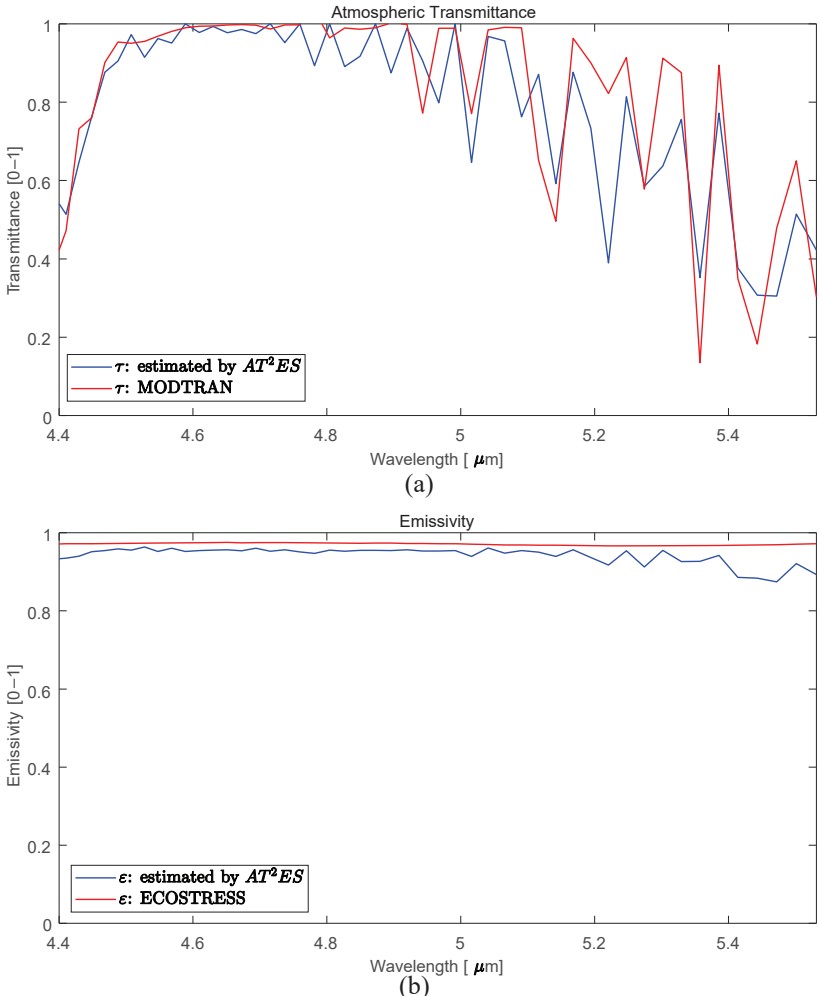

**Figure 24.** Grass region: Comparison of atmospheric transmittance and emissivity estimation by the proposed $AT^2ES$: (**a**) spectral atmospheric transmittance comparison with MODTRAN, and (**b**) spectral emissivity comparison with the ECOSTRESS library.

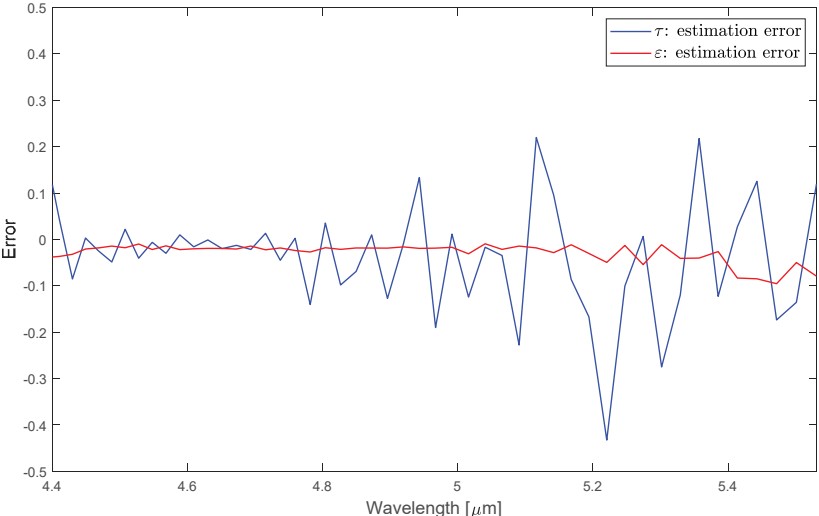

**Figure 25.** Grass region: Estimation error of spectral atmospheric transmittance and emissivity by the proposed $AT^2ES$.

In an asphalt region, the estimated $T_{tg}$ was 41.4 °C. Ground truth for asphalt temperature was hard to measure due to the bumpy structure. In general, solar radiance energy

(visible/near IR) is converted to long wave thermal energy, and the asphalt temperature is higher than the air temperature. FTIR imaging was done at 13:39 h on 21 May 2020.

Figure 26 shows the estimated spectral atmospheric transmittance and emissivity, compared with MODTRAN and the ECOSTRESS grass library. The MODTRAN simulation was the same as the grass experiment. Note that $AT^2ES$ can estimate spectral atmospheric transmittance quite accurately, as shown in Figure 26a. In the emissivity comparison, sample ID 0095UUUASP (Paving Asphalt) from the ECOSTRESS spectral library was chosen because it was most similar to our asphalt region. As shown in Figure 26b, $AT^2ES$ estimated a similar emissivity profile considering complex asphalt composition, but with some emissivity offset. Figure 27 visualizes the spectral estimation error of $\tau(\lambda)$, $\varepsilon(\lambda)$. The MAE for emissivity was 0.041. Note that large errors were generated around low atmospheric transmittance bands in the grass experiment.

Interestingly, if we add an object temperature offset of 2 °C to $T_{tg}$, the estimated emissivity moves upward as shown in Figure 28a, with the same emissivity profile shape. Figure 28b shows the estimation error profile of atmospheric transmittance and emissivity. The MAE of emissivity was reduced to 0.023 from 0.041. From this additional test, the proposed $AT^2ES$ estimated a lower object temperature for low emissivity material, which leads to an emissivity profile with an offset. This is a future research direction to improve $AT^2ES$ for low-emissivity objects.

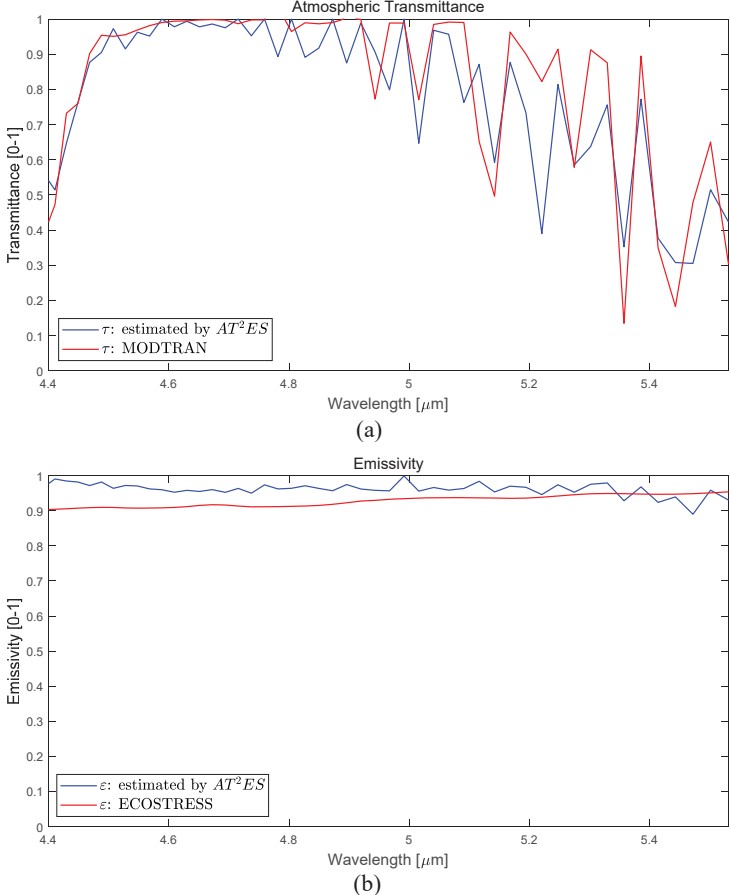

**Figure 26.** Asphalt region: Comparison of atmospheric transmittance and emissivity estimation by the proposed $AT^2ES$ and MODTRAN: (**a**) spectral atmospheric transmittance, and (**b**) spectral emissivity.

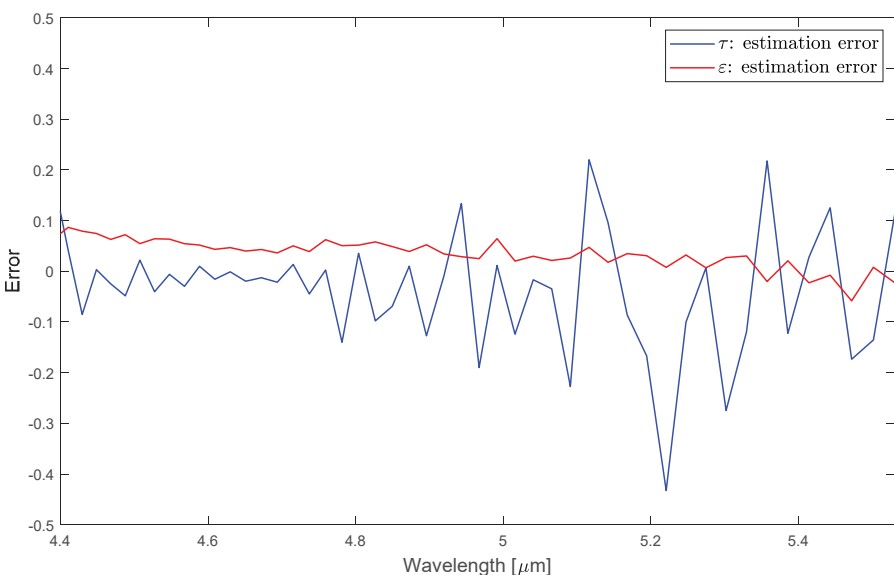

**Figure 27.** Asphalt region: Estimation error in spectral atmospheric transmittance and emissivity by the proposed $AT^2ES$.

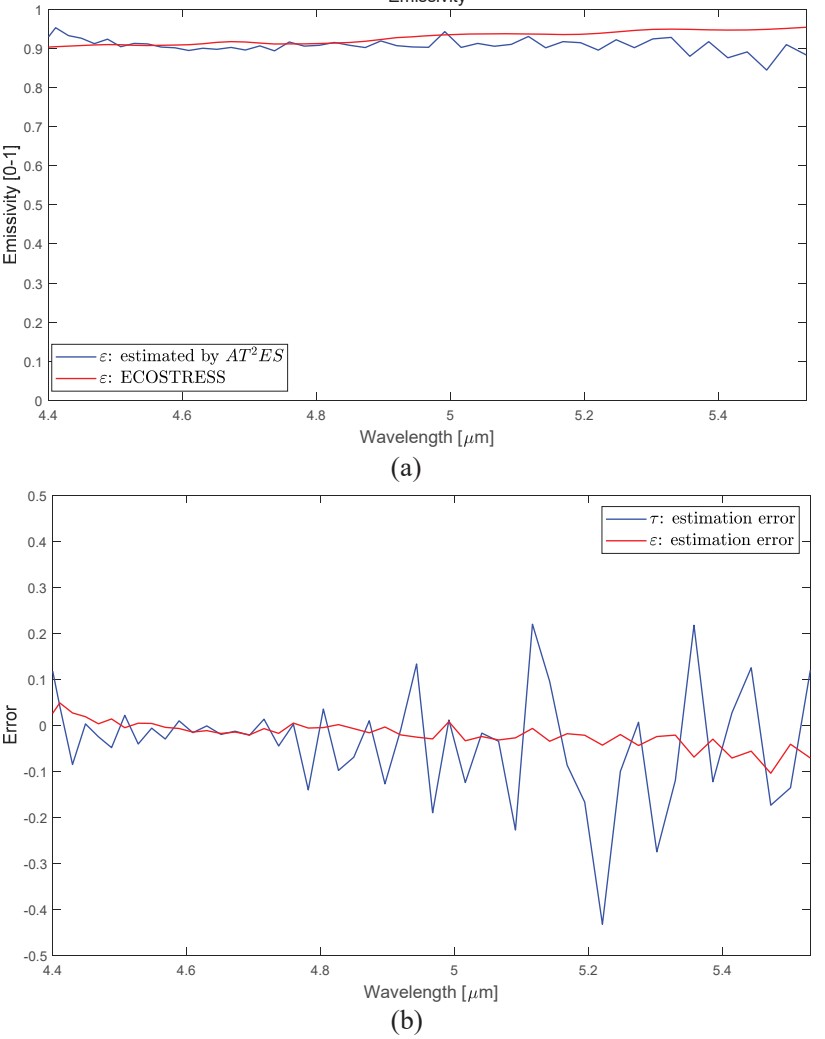

**Figure 28.** Asphalt region: Control of the emissivity offset by adding an object temperature offset: (**a**) spectral emissivity, and (**b**) estimation error in spectral atmospheric transmittance and emissivity.

## 4. Conclusions

Temperature emissivity separation (TES) is an important research topic in a remote-sensing society. Most approaches use atmospheric correction to remove atmospheric transmittance, downwelling, and upwelling generated from MODTRAN. However, online atmospheric information changes from time to time and region by region. This paper presents $AT^2ES$, a novel method to separate atmospheric transmittance, temperature, and emissivity simultaneously without the aid of an offline MODTRAN simulation.

The key idea is based on radiometry transfer properties in the upper MWIR band (4.2–5.6 μm) where there are negligible downweilling and solar upwelling components (1–4%) with a high emissivity surface (above 0.9) at a 100 m distance. From the proposed approximate RTE, the $AT^2ES$ algorithm can separate four parameters simultaneously. Air temperature is extracted from the brightness temperature in the $CO_2$ absorption band (4.20–4.35 μm). The object surface temperature is obtained by applying the max operation to the brightness temperature, except the $CO_2$ absorption band. Given some observed spectral radiance samples and an object temperature, data regression of object blackbody radiance and the observed radiance can provide the slope and intercept. In particular, spectral atmospheric transmittance is separated using the y-intercept and air blackbody radiance. The separated atmospheric transmittance is the same for all the samples, but each sample has different emissivity with the same atmospheric transmittance. Therefore, each spectral emissivity is calculated using the separated parameters. The average operation can provide a representative spectral emissivity profile for a certain region.

The first experiment using synthetic spectra provided the effects of noise in the four parameters. Object surface temperature error directly affects spectral emissivity and air temperature. The air temperature error affects atmospheric transmittance, object temperature, and air temperature. Atmospheric transtmittance error directly affects the estimation of atmospheric transmittance. The object emissivity error also affects atmospheric transmittance. The second experiment was based on an outdoor dataset to check the feasibility of the proposed $AT^2ES$. In grass region samples, the separated temperature parameters were very close to the measured temperatures. Separated spectral atmospheric temperature and emissivity were similar to the profiles in MODTRAN. This is due to the high emissivity of grass regions. In an asphalt region, the estimated emissivity was rather higher than in the ECOSTRESS profile due to lower object temperature estimation. If the object temperature was increased by 2 °C, spectral emissivity was consistent with the spectral library. Therefore, a future research direction is to find an improved $AT^2ES$ method for low emissivity materials.

**Author Contributions:** The contributions were distributed between authors as follows: S.K. (Sungho Kim) wrote the text of the manuscript, programmed the hyperspectral $AT^2ES$ method using upper MWIR-FTIR data. J.S. and S.K. (Sunho Kim) provided the midwave infrared hyperspectral database, operational scenario, performed the in-depth discussion of the related literature, and confirmed the accuracy experiments that are exclusive to this paper. All authors have read and agreed to the published version of the manuscript.

**Funding:** This research was funded by ADD grant number UE191095FD, 2021 Yeungnam University Research Grants, and NRF (NRF-2018R1D1A3B07049069).

**Acknowledgments:** This study was supported by the Agency for Defense Development (UE191095FD). This work was supported by the 2021 Yeungnam University Research Grants. This research was also supported by Basic Science Research Program through the National Research Foundation of Korea (NRF) funded by the Ministry of Education (NRF-2018R1D1A3B07049069).

**Conflicts of Interest:** The authors declare no conflict of interest.

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
