# Peer review of "AT2ES: Simultaneous Atmospheric Transmittance-Temperature-Emissivity Separation Using Online Upper Midwave Infrared Hyperspectral Images"

_remotesensing, doi:10.3390/rs13071249_

Round 1

Reviewer 1 Report

This paper presents a novel method for atmospheric transmittance-temperature-emissivity separation (AT2ES) using online midwave infrared hyperspectral images. Also, this method was verified using two experiments, and the results seemed to be reasonable. However, some revisions are needed.

Comments:

  1. The daytime radiance measured at the top of the atmosphere in MIR region includes the direct solar radiation reflected by the surface. The reflectance is a bidirectional reflectivity of the surface. In equation 1, the authors equated the bidirectional reflectance to 1 minus emissivity, will this assumption make some errors?
  2. I think the equation 3 is wrong (see equation 3, equation 5, and line 158), please check it.
  3. Line 86-88, authors neglected the surface-reflected downwelling thermal radiance and path reflectance-solar radiance. Although the proportion of these two kinds of radiation energy in the total energy is not high, how much error does this assumption make in the results.
  4. Figure 7, the transmittance is 0.992, when the object distance is 100 m. What are the atmospheric parameters the author simulated in Figure 7, such as atmospheric water vapor content, etc. In addition, can the authors give an average atmospheric transmittance? I wonder if the transmittance would be so high when the atmospheric moisture content was higher.
  5. Line 180, the , why is so small compared to .
  6. In the second experiment, did the authors consider the problem of view zenith angle? In my opinion, it was not vertical observation in the experiment.
  7. Line 228, why the grass temperature is almost the same as air temperature? What about soil temperature or rock temperature? And what is the definition of air temperature, is it the temperature of the bottom of the atmospheric profile or the equivalent temperature of the atmospheric profile.

Author Response

Dear reviewer,

The revision note file is attached  below. 

Thank you very much.

Reviewer 2 Report

This paper presents a novel approach, to estimate atmospheric transmittance, temperature, and emissivity from MWIR hyperspectral imager, with newly designed radiative transfer equation approximation and estimation flow. The proposed method does not require prior knowledge and pre-processing stage, and both synthetic and real hyperspectral data experiments demonstrate efficient results.

This reviewer thinks the proposed idea is good and worth to be published, but there are still some issues this reviewer thinks could be improved. Below are some detail comments and suggestions for improvement from this reviewer:

  1. The notations in Figure 1 and Equation (1) are quite confused. Please explain the notations in Figure 1, especially the LS, LT, Ls, and Lt.
  2. In Equation (3) and (5), there is a ‘)’ lost before the LBB.
  3. The structure of experimental section is not well-organized. This reviewer suggests authors to divide this section into 2 sections or 2 sub-sections, such as synthetic datasets experiments and real datasets experiments.
  4. In both synthetic and real hyperspectral data experiments, this reviewer suggests authors to document R2 in the stage of estimating slope and intercept with regression, to illustrate the fitting performance of the linear model.
  5. In figure 20, this reviewer suggests it will be better if authors can provide a band image from the hyperspectral imager of the test field.
  6. Please check the style of references, the cited journals should be abbreviated.

Author Response

(The authors gave the same response as above.)

Round 2

Reviewer 1 Report

The authors have revised all of the comments.